# Changes in the fine-scale genetic structure of Finland through the 20[th] century

Sini Kerminen[1], Nicola Cerioli[2], Darius Pacauskas[2,3], Aki S. Havulinna[1,4], Markus Perola[4], Pekka Jousilahti[4], Veikko Salomaa[4], Mark J. Daly[1,5], Rupesh Vyas[2], Samuli Ripatti[1,5,6], Matti Pirinen[1,6,7]*

1 Institute for Molecular Medicine Finland, Helsinki Institute of Life Science, University of Helsinki, Helsinki, Finland, 2 Department of Media Design, School of Arts, Design and Architecture, Aalto University, Espoo, Finland, 3 Autovista Group, Helsinki, Finland, 4 Finnish Institute for Health and Welfare, Helsinki, Finland, 5 Broad Institute of MIT and Harvard University, Cambridge, Massachusetts, United States of America, 6 Department of Public Health, University of Helsinki, Helsinki, Finland, 7 Department of Mathematics and Statistics, University of Helsinki, Helsinki, Finland

* matti.pirinen@helsinki.fi

**Data Availability Statement:** All data underlying the findings are available via THL Biobank (https://thl.fi/en/web/thl-biobank/for-researchers) for research projects that are of high scientific quality and impact, are ethically conducted, and that

## Abstract

Information about individual-level genetic ancestry is central to population genetics, forensics and genomic medicine. So far, studies have typically considered genetic ancestry on a broad continental level, and there is much less understanding of how more detailed genetic ancestry profiles can be generated and how accurate and reliable they are. Here, we assess these questions by developing a framework for individual-level ancestry estimation within a single European country, Finland, and we apply the framework to track changes in the fine-scale genetic structure throughout the 20[th] century. We estimate the genetic ancestry for 18,463 individuals from the National FINRISK Study with respect to up to 10 genetically and geographically motivated Finnish reference groups and illustrate the annual changes in the fine-scale genetic structure over the decades from 1920s to 1980s for 12 geographic regions of Finland. We detected major changes after a sudden, internal migration related to World War II from the region of ceded Karelia to the other parts of the country as well as the effect of urbanization starting from the 1950s. We also show that while the level of genetic heterogeneity in general increases towards the present day, its rate of change has considerable differences between the regions. To our knowledge, this is the first study that estimates annual changes in the fine-scale ancestry profiles within a relatively homogeneous European country and demonstrates how such information captures a detailed spatial and temporal history of a population. We provide an interactive website for the general public to examine our results.

## Author summary

We have inherited our genomes from our parents, who, in turn, inherited their genomes from their parents, etc. Hence, a comparison between genomes of present day individuals reveals genetic population structure due to the varying levels of genetic relatedness among

correspond with the research areas of THL Biobank. The script for generating haplotypes for simulations: https://github.com/sinikerm/OffspringSimulator.

**Funding:** This work was supported by the Academy of Finland (https://www.aka.fi/en/) (Grants 288509 and 319181 to M.Pi.), the Academy of Finland Center of Excellence in Complex Disease Genetics (312076 to M.Pi; 312062 to S.R.), by the Sigrid Juselius Foundation (https://sigridjuselius.fi/en/) (to M.Pi and S.R.) by University of Helsinki (https://www.helsinki.fi/en) HiLIFE Fellow and Grand Challenge grants (M.Pi. and S.R.), by the Finnish Foundation for Cardiovascular Research (https://www.sydantutkimussaatio.fi/en) (S.R.) and by Professor Package A80202-921072-Vyas from Aalto University (https://www.aalto.fi/en) to R.V. The funders had no role in study design, data collection and analysis, decision to publish, or preparation of the manuscript.

**Competing interests:** I have read the journal's policy and the authors of this manuscript have the following competing interests: V.S. has received honoraria from Novo Nordisk and Sanofi for consultations. He also has ongoing research collaboration with Bayer Ltd. (All unrelated to the present study).

the individuals. We have utilized over 18,000 Finnish samples to characterize the fine-scale genetic population structure in Finland starting from a binary East-West division and ending up with 10 Finnish source populations. Furthermore, we have applied the resulting ancestry information to generate records of how the population structure has evolved each year between 1923 and 1987 in 12 geographical regions of Finland. For example, the war-related evacuation of Karelians from Southeast Finland to other parts of the country show up as a clear, sudden increase in the Evacuated ancestry elsewhere in Finland between 1939 and 1945. Additionally, different regions of Finland show very different levels of genetic mixing in 1900s, from little mixed regions like Ostrobothnia to highly mixed regions like Southwestern Finland. To distribute the results among general public, we provide an interactive website for browsing the municipality and region-level genetic ancestry profiles at https://geneviz.aalto.fi/genetic_ancestry_finland/

## Introduction

A genetic ancestry profile of an individual tells which proportion of the individual's genome originates from each of the available reference populations. Such a profile provides a unique view to the individual's personal history, is a crucial component in emerging genomic medicine[1] and is central for forensic genetics[2]. Therefore, it is of great interest to determine how detailed an ancestry profile we are able to robustly generate by current data resources and computational methods.

A variety of methods have been proposed to estimate genetic structure and individual-level ancestry[3–5]. The most popular methods, such as f-statistics (ADMIXTOOLS)[6], principal component analysis (PCA) (e.g. EIGENSOFT)[7] or ADMIXTURE[8] are based on genotype frequencies of independent variants and explore ancestry on a time-scale of hundreds of generations. Thus, these methods may not be optimal for the studies of recent past or fine-scale relationships between subpopulations. In contrast, current haplotype-based methods, such as FineSTRUCTURE[9], detect population genetic differences in striking detail, for example, in Britain and Ireland [10–13], Japan[14], Italy[15] and France[16], and provide better resolution for genetic structure in the recent past. So far, the haplotype-based methods, such as GLOBE-TROTTER[17] and SOURCEFIND[18], have been applied to estimate ancestry and date admixture from relatively broad geographic areas, for example, in Europe[11, 13, 19, 20], Africa[21–23] and Eurasia[24, 25]. Consequently, there remains limited information about the accuracy and robustness of individual-level ancestry estimation using fine-scale source populations. In this study, we assess these questions within a single European country, Finland.

The Finnish population has been widely used in human genetic studies[26] and due to an ongoing large-scale biobank collection of 500,000 samples by the FinnGen project (https://www.finngen.fi/en), Finland will likely remain as one of the most accessible and best characterized populations for future research in human genetics. We have previously characterized the fine-scale genetic structure of Finland that prevailed during the first half of the 20<sup>th</sup> century [27, 28], before urbanization and the large-scale migration events related to World War II. During and after the war (from 1939 to 1945), over 400,000 (11% of total population) individuals left their homes as Finland lost parts of its eastern territories to the Soviet Union (Fig 1). Almost 70% of the evacuees were relocated to the southern or western parts of Finland while around 25% were relocated to Eastern Finland and 5% were assigned to the northern parts of the country[29]. Later, starting from 1950s, urbanization has shaped the population distribution within Finland[30] and now the biggest cities locate in southern and western parts of the

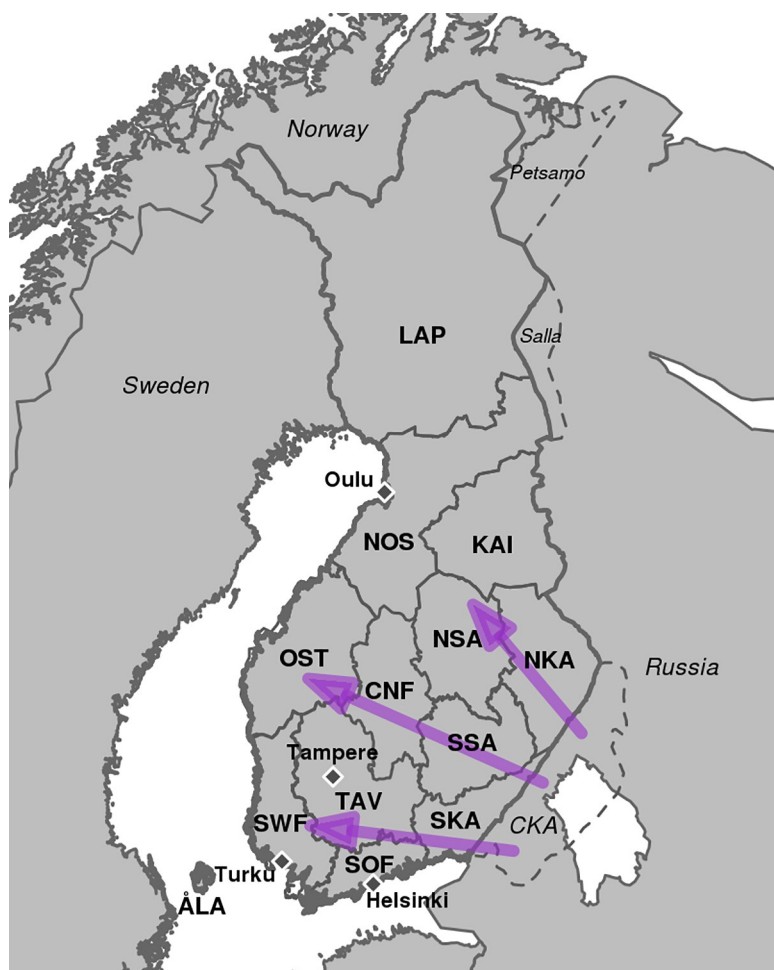

**Fig 1. Map of Finland with the study regions.** Map of Finland showing historically motivated regions that we use to track changes in genetic ancestry. Dotted lines show the regions ceded to the Soviet Union (Ceded Karelia (CKA), Petsamo, and Salla) after World War II. Today these three regions are part of Russia. Purple arrows show the main patterns how evacuees from CKA were relocated to other parts of Finland. Diamonds mark the four largest metropolitan areas in Finland: Helsinki, Turku, Tampere and Oulu. ÅLA: Åland islands, SOF: Southern Finland. SWF: Southwestern Finland, TAV: Tavastia, SKA: South Karelia, CKA Ceded Karelia, NKA: North Karelia, SSA: South Savo, NSA: North Savo, CNF: Central Finland, OST: Ostrobothnia, NOS: North Ostrobothnia, KAI: Kainuu, LAP: Lapland.

country (Fig 1). These two well-documented migration events together with over 18,000 samples from the National FINRISK Study with genotype, birth year (range 1923–1987) and birth place information at municipality level, provide us with an opportunity to study how demographic changes are reflected in genetic population structure at an unprecedented spatial and temporal detail.

In this study, we will first define genetically and geographically homogeneous reference groups within Finland at three levels of detail and evaluate their ability to detect ancestry from various geographic regions via simulations. Second, we estimate the genetic ancestry profiles of over 18,000 Finnish individuals with respect to the reference groups using a haplotype-based method SOURCEFIND[18]. By stratifying the ancestry profiles based on birth year and birth place, we track the annual changes in the fine-scale genetic structure of Finland through the 20th century. The results demonstrate high accuracy to detect both the fine-scale

individual-level ancestry profiles, as well as the sudden internal migration of the World War II evacuees and the region-specific rates in genetic diversification due to urbanization in Finland.

To our knowledge, this is the first study to track geographically the annual genetic contributions of subpopulations within a single European country. Our work concretely demonstrates continuous genetic mixing within current human populations. To convey our results to the general public, we provide a website for interactive examination of our results.

## Results

To create and test a framework for individual-level ancestry estimation within Finland, we apply haplotype-based computational methods, ChromoPainter, FineSTRUCTURE[9] and SOURCEFIND[18] on geographically and genetically informed data set from the National FINRISK Study[31]. The general validity of these computational methods have been evaluated previously (e.g. [9, 10] for ChromoPainter/FineSTRUCTURE and [18] for SOURCEFIND). Therefore, we focus on the particular application of these methods to create a reliable and easily interpretable individual-level ancestry estimation framework within Finland. We will first introduce a procedure to identify suitable reference groups, then we use simulations to test the performance of our reference groups in detecting ancestry, and finally apply them to estimate the ancestry of 18,463 FINRISK samples to characterize the fine-scale changes in the genetic structure of Finland through the 20<sup>th</sup> century. S1 Fig describes the workflow of the study.

### Identifying reference groups for ancestry estimation within Finland

Our first goal was to define geographically and genetically motivated reference groups within Finland to serve as a basis for robust individual-level ancestry estimation. We did this on several levels of detail, starting from the binary division between eastern and western Finland, and further refining the sources of genetic ancestry to 6 and finally to 10 reference groups within Finland, as explained below. We refer to these three sets of reference groups with the term "refset" as an abbreviation for "reference set".

We used the municipalities of birth of parents to identify 2,741 geographically precisely located and evenly distributed unrelated individuals (see Materials and methods for detailed description). We call these individuals "reference candidates" as we later further filtered them down to form our final reference groups. The refence candidates were analyzed with Chromo-Painter and FineSTRUCTURE to identify groups that represented the fine-scale population structure in Finland (Materials and methods). The fine-scale population structure can be studied by cutting the hierarchical tree from FineSTRUCTURE (FS-tree) at different levels. In S3 Fig, we confirm that our results closely match with previous results on the fine-scale structure in Finland[27, 32] showing a main division between the east and the west that further splits into dozens of geographically well-defined and fairly equal sized subpopulations throughout the country.

To identify statistically separable reference groups, we evaluated the fine-scale populations with the following procedure:

1. **Starting level**
   Choose K, the starting number of FineSTRUCTURE populations from the FS-tree.

2. **Initial ancestry**
   Estimate the genetic ancestry of the reference candidates with respect to the K populations using SOURCEFIND.

3. **Identity proportions**
   For each of the K populations, calculate the population's *identity proportion* as the average

proportion of ancestry in that particular population across the individuals assigned to that population by FineSTRUCTURE.

4. **Population exclusions**
   Exclude the populations with low identity proportions ($< 50\%$) from the reference candidates, decrease K accordingly, and repeat from step 2. If no population is excluded, proceed to step 5.

5. **Candidate exclusions**
   Exclude the reference candidates who show low levels of ancestry from the population they were assigned to by FineSTRUCTURE. (Thresholds used either $< 70\%$ or $< 95\%$.)

6. **Geographic outliers**
   Exclude possible geographic outliers manually (S4 Fig).

As the East-West division in genetic structure is relatively strong in Finland[33], we first focused on the top level of the FS-tree which divided our reference candidates into 926 western and 1,815 eastern individuals. Both populations showed a high identity proportion ($> 87\%$), and we excluded the reference candidates whose both eastern and western ancestry components were below 95% or who were geographically located on the opposite part of Finland compared to their dominant genetic component (S4 Fig). A comparison between Figs 2A and S3A shows that this procedure excluded reference candidates from the borders of the two populations and resulted in geographically more tightly defined reference groups. The final sample sizes of the reference groups of our refset 2 were 497 in western and 975 in eastern Finland (Fig 2A).

To test whether we can detect more detailed ancestry within Finland, we continued by considering the FS-tree of the reference candidates at the level of the first 15 populations. The 15 populations showed differing identity proportions (S5A Fig), suggesting that some populations were more mixed and/or so closely related to some other populations that they could not be

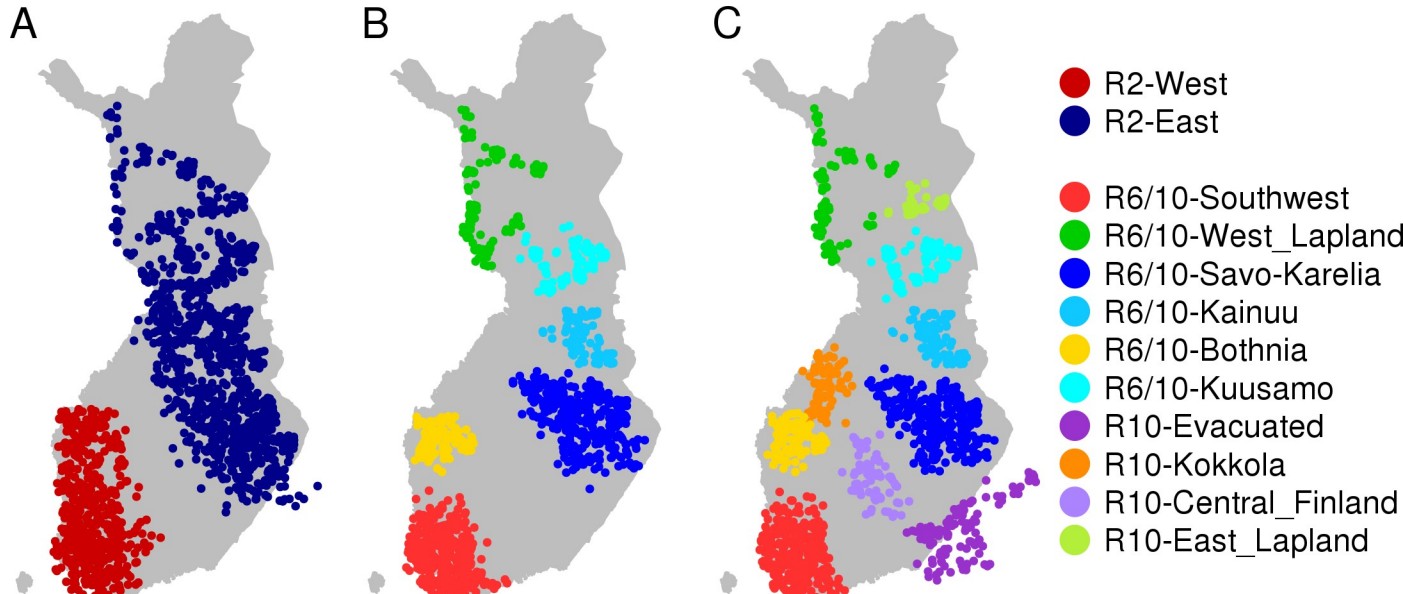

**Fig 2. Geographic location of reference individuals.** The Geographic location of the reference individuals in the reference groups of A) refset 2 (n = 1,472), B) refset 6 (n = 1,026) and C) refset 10 (n = 1,236) marked with colors. The names of the reference groups are shown on right. The locations were determined as the mean of the parents' municipalities of birth.

reliably distinguished from the other populations. To ensure robust ancestry estimation, we excluded the populations whose identity proportion was below 50%. This left us with 10 populations. We reran the ancestry estimation for the remaining reference candidates using the 10 populations as reference groups and recalculated the identity proportions. The 10 populations were clearly split into two groups: 6 populations with identity proportion around 80% and 4 populations with identity proportion around 70% (S5B Fig). In what follows, we will study the detailed ancestry estimation using both the more clearly distinguishable 6 populations (refset 6), and also the 10 populations (refset 10) that provide a larger geographic coverage (Fig 2B and 2C). The reference individuals at refset 10 are included in their corresponding population at refset 6 (except for the four refset 10 populations that were not present at refset 6). Furthermore, from the reference groups in refset 6 or in refset 10, we excluded the individuals who showed an identity proportion below 70%, as well as 11 refset-6 and 16 refset-10 individuals who were geographically outside of their own population (S4 Fig). S6 Table shows the number of excluded or included reference candidates. The distributions of birth years for the reference groups are shown in S6 Fig and S4 Table.

To avoid confusion between the names of the geographic regions and the names of the genetic ancestry groups we use the following conventions. For the names of the geographic regions, we use their full names or the three letter abbreviations (e.g. North Karelia or NKA) given in Fig 1. When referring to the genetic ancestry from a reference group, we use the prefix *R* with the index of a refset and a name describing the geographic location of the reference group, as given in Fig 2 (e.g. R2-East or R10-Evacuated). Finally, when we evaluate the framework through simulations, we refer to the "ancestor groups" similarly to the reference groups but using the prefix *A* instead of *R* (e.g. A-East), as described in S9 Fig.

Table 1 shows that genetically the most distant reference groups are R10-Bothnia and R10-East_Lapland with $F_{ST}$ ~0.007 (computed with EIGENSOFT[7]). The smallest difference ($F_{ST}$ ~0.002) appears between R10-Evacuated and R10-Southwest, R10-Central_Finland and R10-Savo-Karelia highlighting the status of R10-Evacuated between the east and the west. Also, R10-Savo-Karelia and R10-Kainuu are closely related groups.

## Identifiability of ancestry from reference groups

We tested the identifiability of ancestry from different reference groups using simulations where $2^G$ individuals were sampled to represent the ancestors from G generations back in time (G varied between 1 and 5). We simulated the meioses within these ancestors, and within their subsequent descendants in generations G-1, G-2, . . .,1, to determine the genotypes of the target individual at generation 0. The ancestry of the target individual was then estimated and compared to the expected ancestry groups of the sampled ancestors based on their geographic and genetic origin (see S7 Fig for a schematic representation).

## Refset 2

First, we tested the identifiability of eastern vs. western ancestry by SOURCEFIND using our refset 2. We simulated individuals using four scenarios where either all ancestors came from the same region (*All-West*, *All-East*) or one ancestor came from A-East and the remaining ancestors came from A-West (*Almost-West*) or vice versa (*Almost-East*). The set of ancestor candidates was disjoint from our reference individuals, and they were chosen by their parents' geographic location and a standard principal component analysis (S8 Fig), with their locations shown in S9A Fig. For each scenario, and for each number of generations from 1 to 5, we randomly chose the ancestors (2, 4, 8, 16 or 32 ancestors depending on the number of generations), simulated the meioses from the ancestors to the target individual by sampling a

**Table 1. Pairwise-$F_{ST}$ values ($\times 10^5$) between reference groups of refset 10 (lower triangular) and the corresponding standard errors (upper triangular).**

| | R10-Southwest | R10-Bothnia | R10-Kokkola | R10-Evacuated | R10-Central_Finland | R10-Savo-Karelia | R10-Kainuu | R10-Kuusamo | R10-West_Lapland | R10-East_Lapland |
|---|---|---|---|---|---|---|---|---|---|---|
| R10-Southwest | - | 5 | 5 | 5 | 6 | 5 | 8 | 8 | 6 | 12 |
| R10-Bothnia | 251 | - | 8 | 7 | 8 | 8 | 10 | 9 | 8 | 14 |
| R10-Kokkola | 267 | 338 | - | 7 | 9 | 7 | 10 | 10 | 9 | 14 |
| R10-Evacuated | 238 | 371 | 343 | - | 6 | 4 | 7 | 7 | 7 | 11 |
| R10-Central_Finland | 268 | 397 | 367 | 160 | - | 6 | 8 | 9 | 9 | 14 |
| R10-Savo-Karelia | 404 | 498 | 405 | 185 | 226 | - | 5 | 6 | 7 | 11 |
| R10-Kainuu | 546 | 637 | 542 | 362 | 380 | 219 | - | 8 | 10 | 13 |
| R10-Kuusamo | 538 | 639 | 544 | 392 | 423 | 302 | 363 | - | 9 | 12 |
| R10-West_Lapland | 369 | 505 | 472 | 415 | 454 | 497 | 632 | 602 | - | 13 |
| R10-East_Lapland | 557 | 661 | 582 | 503 | 545 | 514 | 618 | 538 | 497 | - |

recombination process in the ancestral haplotypes (Materials and methods), and estimated the ancestry profile of the target individual. Fig 3 shows the average ancestry profiles of 20 simulated individuals (S10 Fig shows all the 20 individual ancestry profiles) and demonstrates that individuals with a single origin (*All-West* or *All-East*) show high levels of ancestry (>96%) from the expected reference group. For the individuals with mixed background (*Almost-West*, *Almost-East*), the estimated ancestry in the minor reference group decreases approximately as expected, that is, as $1/2^G$ when 1 out of $2^G$ ancestors G generations back in time come from the minor reference group. We noticed that we always estimate at least some small proportion (~3%) from both of the refset-2 reference groups, even when all ancestors were chosen from a single ancestor group. While this may well reflect a small but real ancestry proportion in our ancestor candidates, it also suggests that an upper limit of a reliable detection of direct refset-2 ancestry is 4 generations (6.25% of ancestry) rather than 5 generations (3.13% of ancestry) back in time. In S1 Table, we explicitly tested whether we can distinguish when an individual has all ancestors G generations back in time from a single source (e.g. *All-West*) from the case when one of the ancestors is from the different source (e.g. *Almost-West*). The results verify that we can identify correctly the east-west origin of ancestors 4 generations back in time for over 75% of the individuals but that we cannot reliably do the same 5 generations back in time.

## Refsets 6 and 10

To test the accuracy of the detailed ancestry estimation using either refset 6 or refset 10, we simulated individuals with the ancestor candidates shown in S9B Fig. These ancestor

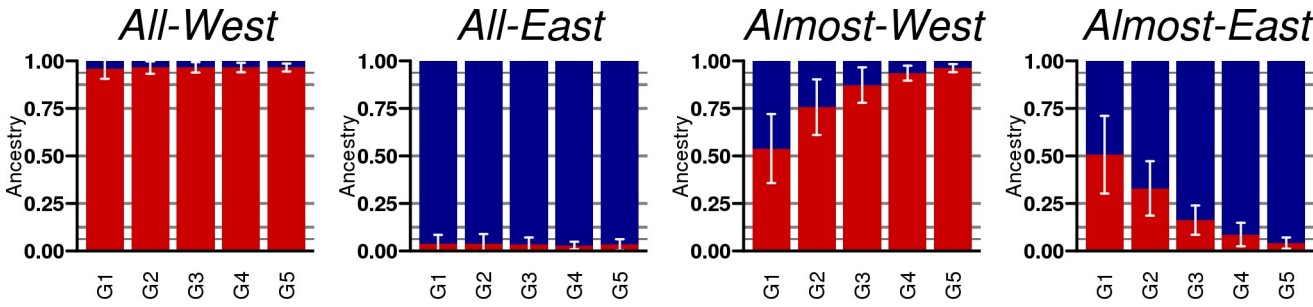

**Fig 3. Average ancestry profiles for simulation scenarios between East and West.** Average ancestry profiles for four simulation scenarios and up to 5 generations back in time (G1 –G5). In *All-West* and *All-East* scenarios all ancestors from G = 1. . .5 generations back in time originate from a single ancestor candidate group, A-West or A-East, respectively. In *Almost-West* and *Almost-East* scenarios a single ancestor, G generations back in time, originates from the other group, A-East or A-West, respectively. Each bar shows the average over 20 simulated individuals (individually shown in S10 Fig) with 95% confidence intervals. The colors denote the ancestry profile with respect to refset 2 (Fig 2A): R2-West in red and R2-East in blue.

candidates had their both parents born within 80km, had similar genetic background based on PCA (S8 Fig), and originated approximately from the same geographical regions as the individuals in the reference groups (Materials and methods). The ancestor candidates covered well the geographic regions of the refset 6 reference groups and, in addition, the region of the R10-Evacuated reference group. For the remaining three reference groups in refset 10 (R10-Central_Finland, R10-Kokkola and R10-East_Lapland), we did not have enough ancestor candidates outside the reference group to perform simulations.

Fig 4A shows the mean of the estimated ancestry for the simulated individuals whose both parents originated from one geographic region (single origin) using refset 2, refset 6 or refset 10 (S12 Fig shows the individual ancestry profiles). For the single origin individuals from A-Southwest, A-Bothnia, A-N_Karelia, A-Kainuu and A-Kuusamo our approach estimated high ancestry proportions (~80%) for their closest reference group at all three refsets. For A-Lapland, however, we estimated more varying ancestry proportions with respect to refset-6 or refset-10, and, on average, only 60% of ancestry was from the reference groups geographically located in Lapland (LAP in Fig 1). A comparison of pairwise-$F_{ST}$ between the pairs of reference groups and groups of ancestor candidates (S2 Table) showed that the pairs of reference-ancestor groups in Lapland were significantly more distant from each other than other geographically close reference-ancestor pairs, suggesting that the lower ancestry estimates are likely due to real genetic differences between the reference groups and the ancestor candidates in Lapland. The individuals simulated with ancestors from the A-Evacuated group showed mixed ancestry with refsets 2 and 6, whereas with refset 10 they showed about 75% ancestry from the R10-Evacuated reference group. This demonstrates that the interpretation of ancestry profiles depends crucially on the reference groups available. In addition, we saw that, with both refset 6 and refset 10, each reference group included in the analysis showed on average at least ~2% contribution in the single-origin simulation settings (S3 Table). We noticed that by

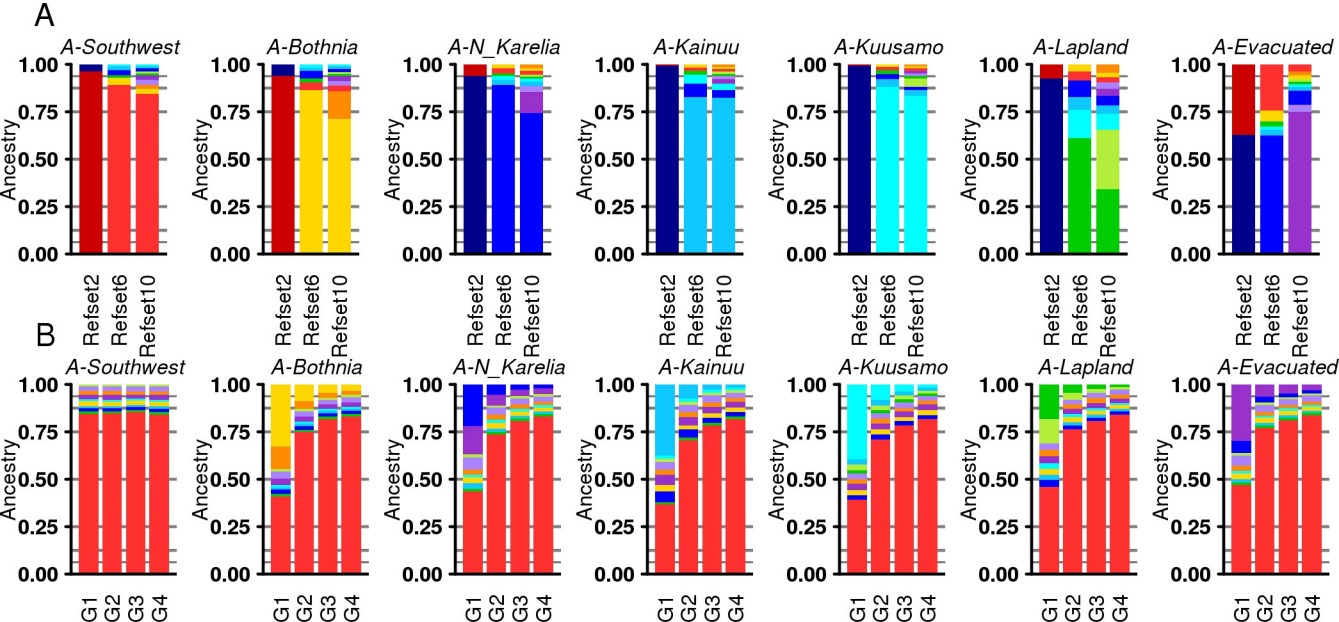

**Fig 4. Average ancestry profiles for detailed simulation scenarios.** The average ancestry for 20 simulated individuals whose parents originate from the ancestor candidate groups described in S9 Fig. Panel A presents individuals whose all ancestors come from one group (single origin) estimated using refset 2, 6 or 10. Panel B presents individuals whose $2^G$-1 ancestors, where G = 1...4 is the number of generations, originate from A-Southwest and 1 ancestor originates from the ancestor group in the title, estimated using refset 10. The colors correspond to the reference groups in Fig 2.

shrinking the ancestry proportions below 5% to zero and by rescaling the remaining ancestry proportions to 100%, the total ancestry from the expected reference groups increased and the average unexpected ancestry proportions reduced considerably (S11 Fig).

To examine how well our approach can detect one ancestor coming from one region while all other ancestors come from another region, we simulated additional generations back in time, analogously to the earlier simulations with refset-2. Fig 4B shows the results when all but one ancestor come from A-Southwest, and the single remaining ancestor originates from one of the seven ancestor groups (see S13 Fig for results for refsets 2 and 6 and when the majority of ancestors come from A-N_Karelia and S14 Fig for individual-level results for the major ancestry component). The results demonstrate that the major source of ancestry is estimated as expected, i.e., corresponds to the geographically closest reference group, for generations 1, 2 and 3 back in time, but remains smaller than expected for generation 4. The minor source of ancestry is always underestimated, and starting from generation 3, the original source of ancestry is not clearly distinguishable from other more geographically distant reference groups.

We observe that when simulating individuals that have one ancestor from A-Southwest and one from A-N_Karelia (Fig 4B), we detect almost 14% ancestry in R10-Evacuated reference group, which is more than expected based on the simulated individuals with a single origin in either A-Southwest or A-N_Karelia, who showed on average about 7% ancestry in R10-Evacuated. This suggests that we may overestimate the R10-Evacuated component for an individual who happens to be mixed between eastern and western Finland, possibly because the R10-Evacuated group itself shows some mixing between R2-East and R2-West (Fig 4A A-Evacuated). This observation needs to be kept in mind later when we interpret the patterns of ancestry from R10-Evacuated in the regions of SWF and NKA. Other pairs of regions do not show similar results of mixing (S15 Fig).

The results averaged over individuals demonstrated that our reference groups are able to accurately detect ancestry all around Finland. With refset 2, we can identify Eastern and Western ancestry up to an accuracy of 6% (4 generations back). With refsets 6 and 10, the major source of ancestry is accurately detected 3 generations back in time and while the proportion of the minor ancestry is underestimated, the source of it can be identified 2 generations back. On the other hand, at the level of individual, the ancestry estimates show increasing variance with more heterogeneous genetic background (S10, S12 and S14 Figs), which makes precise conclusions about genetic ancestry challenging for any one individual. Additionally, our ancestor candidates are likely to be less genetically mixed than an average individual with Finnish ancestry; hence our results do not necessarily directly apply to the Finnish individuals whose ancestors are more mixed.

## Changes in the genetic structure of Finland during the 20<sup>th</sup> century

We then applied our ancestry estimation approach to generate ancestry profiles for 18,463 individuals in the FINRISK Study using the refsets 2, 6, and 10. We also included the reference individuals in the analysis but each reference individual itself was excluded from its reference group when estimating its own ancestry profile.

To study the genetic ancestry patterns across the country, we grouped the individuals into 12 groups based on their region of birth. These 12 regions covered the whole mainland Finland (Fig 1) but not Åland islands (ÅLA). For each region, we averaged the ancestry profiles over individuals' birth years by fitting a local regression (LOESS) curve (Materials and methods). Fig 5 shows the temporal changes in the ancestry profile of six regions using refset 10 (see S17–S19 Figs for all regions with refsets 2, 6 and 10). First, we detected that the regions are often genetically dominated by their geographically closest reference group, but, in many regions,

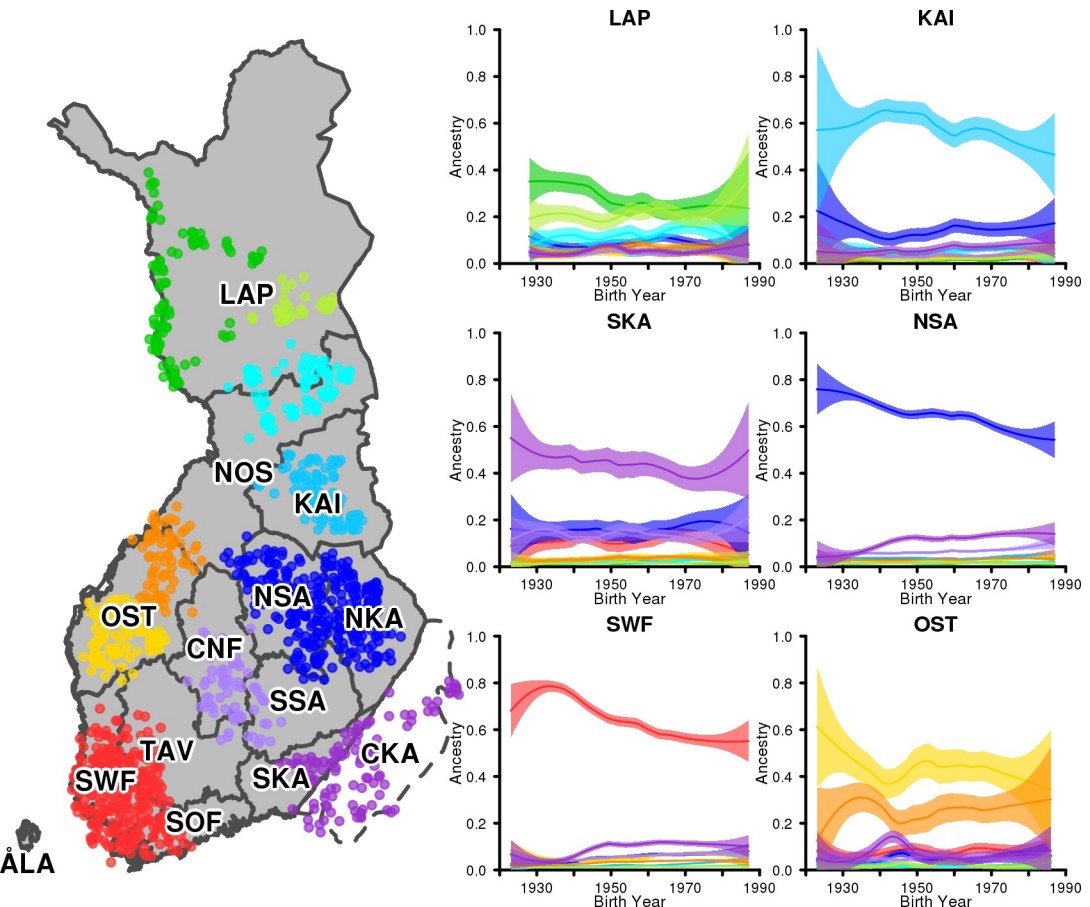

**Fig 5. Development of genetic ancestry through the 20<sup>th</sup> century.** Map names 14 regions and shows the locations of the individuals who form the 10 reference groups. Each panel corresponds to one region and shows 10 curves corresponding to the 10 reference groups. Each curve shows the estimated proportion of ancestry from the reference group, as a function of birth years of individuals from 1923 to 1987. The bands around the curves show 95% confidence intervals.

the proportion of that reference group decreased towards the present, indicating increasing levels of genetic mixing. The most dramatic changes occurred in SOF (23 percentage point decrease in the closest reference group between 1930 and 1980), SWF (22 p.p), TAV (28 p.p) and CNF (21 p.p) while SKA, KAI and NOS did not show considerable changes in their dominant components. The same trends could be detected also with respect to refset 2 (S17 Fig) or refset 6 (S18 Fig).

## World War II evacuees from Ceded Karelia

We identified also some rapid changes which, in some cases, can be dated with an accuracy of one year. The most prominent example is the rapid increase in the ancestry of the R10-Evacuated in SWF and in TAV in the 1940s, which is a result of the massive evacuation and relocation caused by World War II. Figs 6 and S20 show that the increase in R10-Evacuated ancestry could be detected in all regions except in SKA (whose dominant group was already closely genetically related to R10-Evacuated) and in KAI, that, according to the historical records, did not noticeably gain evacuees[29]. Moreover, we could detect widely varying regional patterns of the subsequent movements of the evacuees after the war (Fig 6). In many regions, such as in SWF, TAV and NSA, the increase of the R10-Evacuated ancestry in 1940s was followed by a

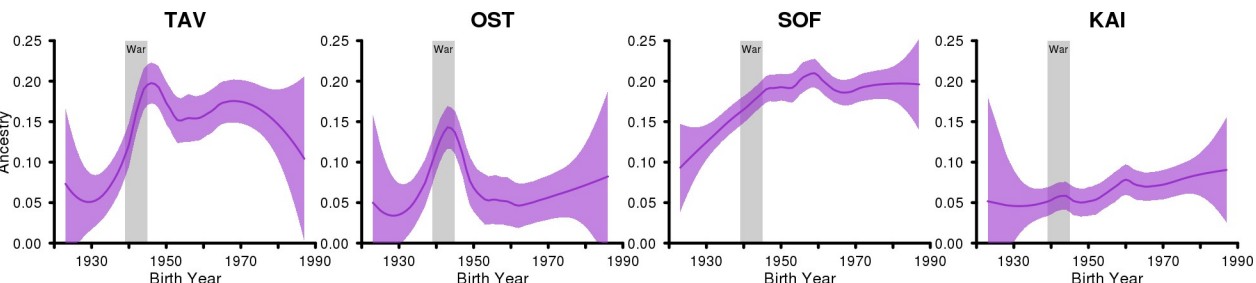

**Fig 6. Regional development of genetic ancestry from R10-Evacuated reference group.** The development of the genetic ancestry from the R10-Evacuated reference group on four regions, Tavastia (TAV), Ostrobothnia (OST), Southern Finland (SOF) and Kainuu (KAI) suggests very different migration patterns of war-time evacuees in different parts of Finland during the war (1939–1945) and after the war.

constant proportion of that ancestry in the following decades. A striking exception is OST, where we detected a rapid decrease in the R10-Evacuated ancestry after 1945, shrinking almost to its pre-war level already by 1950. This suggests that the evacuees did not settle in OST. This observation is supported by the historical records which show that while in 1944, 28% of the evacuees were located in OST, in 1950, the proportion was only 9% (corresponding to 5% of the total population of OST, Table 1 in [29]). A completely opposite pattern is present in SOF, where we detected a continuous increase in the R10-Evacuated ancestry until 1960. This is also supported by the historical records which show that while in 1945 about 17% of the evacuees were located in SOF, by 1960 the proportion had increased to 25%[29].

We also compared the estimated genetic ancestry proportions of R10-Evacuated to the proportion of the Karelian evacuees in different regions in 1950 (Table 2). The comparison is restricted to the regions of SOF, SWF, TAV, OST, NOS+KAI and LAP because, for these regions, the administrative borders have remained fairly stable over the years and their dominant genetic group is distinguishable from the R10-Evacuated ancestry. The comparison showed that the estimated proportion of R10-Evacuated ancestry in individuals born around 1950 was slightly higher in all regions compared to the proportion of evacuees in the regions. This is unlikely to result from a higher fertility rate of the Karelian evacuees compared to the rest of the population since the fertility of evacuees has been reported lower than in the general population[34]. While we cannot fully exclude the possibility of overestimation of R10-Evacuated ancestry in our approach, another likely explanation also remains. The R10-Evacuated does not only capture the ancestry of the actual war-time evacuees but captures also other than war-related migration and ancestry from the nearby region of SKA. For example, in SOF, that contains the capital region of Helsinki and has been a target of incoming migration through

**Table 2. Proportion of Karelian evacuees (from CKA) and the estimated genetic ancestry proportion of R10-Evacuated in different regions in year 1950.**

| Region* | Total population[35] | Number of evacuees[29] | Proportion of evacuees | Genetic ancestry estimate (95% CI) | |
|---|---|---|---|---|---|
| SOF | 667 500 | 70 686 | 0.11 | 0.19 | (0.18–0.21) |
| SWF | 631 000 | 57 834 | 0.09 | 0.11 | (0.10–0.12) |
| TAV | 553 300 | 68 187 | 0.12 | 0.17 | (0.15–0.20) |
| OST | 607 200 | 32 487 | 0.05 | 0.07 | (0.04–0.10) |
| NOS+KAI | 359 800 | 12 495 | 0.03 | 0.07 | (0.07–0.08) |
| LAP | 167 100 | 6 069 | 0.04 | 0.08 | (0.07–0.09) |

*In 1960, CNF was separated from OST (and small parts of CNF also from TAV and SSA).

the 20[th] century, the R10-Evacuated ancestry clearly started to grow already in the 1920s, well before World War II.

### Increase in genetic heterogeneity

To quantify whether the level of heterogeneity of the ancestry profiles changed through the 20th century, we used entropy as a measure of heterogeneity with larger entropy meaning a more heterogeneous profile. With this measure, we do not quantify the absolute heterogeneity in each region, as measured, e.g., by the average inbreeding coefficient of individuals (S23 Fig), but rather we quantify the regional diversity of ancestral backgrounds with respect to our reference groups. Figs 7 and S21 show the regression slopes of heterogeneity of ancestry on birth years in each region. We see that, on average, the genetic profiles diversified towards the present, but also showed notable differences between the regions. The largest changes were observed in SWF and no change at all in OST and NOS. In addition, there was no change in the refset 2 heterogeneity in KAI, CNF, LAP or SKA, suggesting that the diversification in these regions has happened locally between nearby regions rather than more broadly across the main East-West split.

We also compared the rate of change before and after the year 1950 corresponding to the situations before and after the war (S22 Fig). This comparison shows that, on average, the war-related migration had a more diversifying effect than the later events, such as the urbanization starting from 1950s. After the war, increasing heterogeneity is seen in SOF, SWF and TAV that contain the largest metropolitan areas in Western Finland (Helsinki, Turku and Tampere, respectively), and in NKA and NSA in Eastern Finland. In other regions, the urbanization may have gathered people from nearby rural regions into the nearest cities rather than resulted in incoming migration across the country, but traces of such local urbanization within the regions are not detectable in our analysis.

We have made the municipality-level ancestry profiles and the regional changes in ancestry profiles publicly available via an interactive web application.

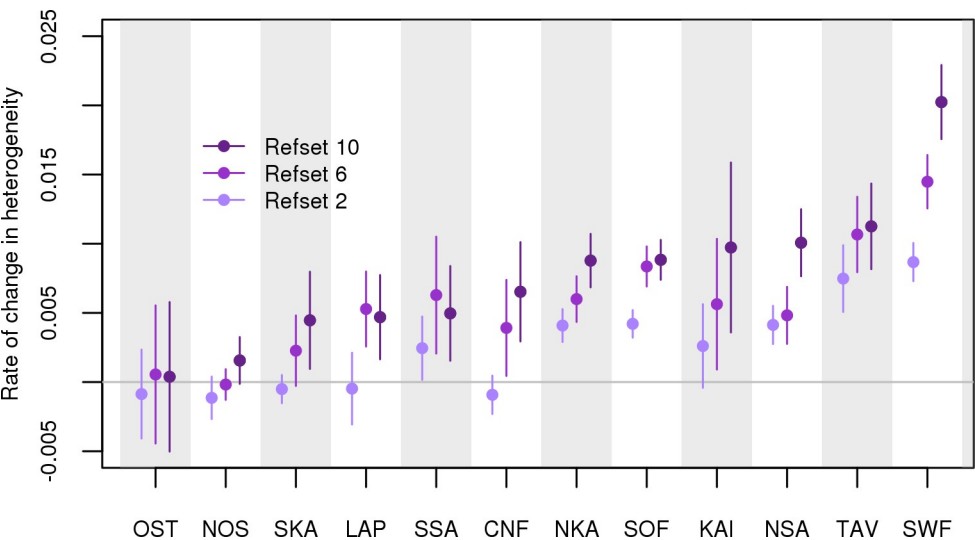

**Fig 7. Rate of change in heterogeneity.** The rate of change in heterogeneity at each region of Finland (see Fig 1) between 1923 and 1987. The rate of change is calculated by regressing the entropy of ancestry profiles of the individuals born in the region on their year of birth. Entropy was calculated with respect to refsets 2, 6, and 10.

## Discussion

Individual-level genetic ancestry information has important scientific applications in genome-wide association studies[36], in genomic medicine[1] and in forensics[2], and it is also of great interest to the general public, as demonstrated by tens of millions of individuals who have taken direct-to-consumer tests motivated by genetic ancestry[37]. The ancestry profiles have so far been estimated at a fairly broad scale by separating ancestry components between continents or countries. In the current phase of rapidly developing biobanks, we have already seen implementations of large-scale genetic studies within individual countries, such as the Finn-Gen project (www.finngen.fi) in Finland that aims to cover 10% of the Finnish population by 2023. Here we have developed and evaluated an ancestry estimation framework for such a targeted, within-country setting. This way we answer the need for accurate ancestry profiles both when implementing genomic medicine via biobank resources and when conveying the potential of large-scale, country-wide genomic resources to the general public. For example, in our work we have been able to track the major demographic events within Finland at the accuracy of one year, and released these results for the general public to browse via an interactive website. While our work uses Finnish data, our analysis framework could be applied to other populations with similar data available.

By grouping individual-level ancestry profiles by birth year and region, we showed how demographic events have affected the fine-scale genetic population structure in Finland. The results verified the prominent effect of the Second World War motivated migration within Finland. While the war-related migration happened within a relatively short period from 1939 to 1945, its diversifying effect on the genetic heterogeneity was larger than the effect of urbanization, that started from the 1950s and has continued to this day. We expect that this result mirrors the pattern of urbanization in Finland: the number of towns and cities has increased rapidly but their size has remained relatively small[30] suggesting that, at least up to 1990s, people have largely moved to their nearby cities rather than migrated long distances across the country. When such a local urbanization event happens within a region dominated by one of our genetic reference groups, it does not change the ancestry profile of the region, and consequently it does not show up in our results.

While our genetic analysis results match well with the known history, the exact interpretation of ancestry and genetic mixing is complicated because it always depends on the available reference groups[38, 39]. An intuitive interpretation of results would require independent reference groups that form a comprehensive collection of all genetic variation present in the region of interest. Unfortunately, this is rarely fully possible in natural populations whose structure is continuous rather than discrete and whose subpopulations are related in various ways to each other. Still, in many cases, useful approximations of discrete reference populations can be found. We did this by first starting with geographically comprehensive and evenly distributed data and by using an unsupervised clustering method (FineSTRUCTURE) to identify statistically separable genetic populations. We got highly consistent results compared to the existing knowledge of the Finnish population structure[27] and we had a good geographic coverage for other parts of mainland Finland except Lapland where the sample size is relatively small and several municipalities cover vast geographic areas. Second, we evaluated the identifiability of the populations with respect to each other by estimating populations' identity proportions. For each population, we defined the identity proportion as the average ancestry proportion of the members of the population in the population itself. A low identity proportion indicates that the population is closely related to one or more other populations included in the analysis. By excluding the populations that showed low identity proportions, we were left with reference groups that showed similar levels of independency and covered well most

parts of Finland. The regions lacking their own reference groups were located near the city of Oulu (NOS) and near the capital region of Helsinki (SOF, TAV), both of which may have gathered recent and genetically heterogeneous migration from elsewhere in Finland. The reference groups were not selected based on the birth year. The mean birth year, that was 1953 across all samples, did not significantly vary between the reference groups, except for R10-Evacuated (mean birth year 1946, Mann-Whitney p-value 4.3e-7) (S6 Fig and S4 Table), whose region of origin was significantly affected by the Second World War.

It is important to acknowledge that the Finnish population today contains a considerable proportion of recent genetic ancestry from outside of Finland as well as from several minority groups who have lived in Finland for centuries (e.g. Roma people) or for millennia (Sami people). A comprehensive ancestry profile of a Finnish individual should therefore include appropriate reference data also from outside of Finland as well as from the minority groups within Finland. Detailed reference data from the neighboring countries would also provide additional information about the relationships between our Finnish reference groups and the populations outside of Finland and open up possibilities to probe history further back in time through genetic analyses.

The ancestor candidates that we used in our simulations were chosen by parental birthplace information and PCA of genetic data, and hence they are expected to be less genetically heterogeneous than an average individual with Finnish ancestry. Consequently, our results about the identifiability of an ancestor with certain genetic background are valid when the ancestor was approximately equally representative of their ancestry group as our ancestor candidates. We have not studied more complex scenarios, where an individual has a considerable proportion of genetic ancestry in a certain reference group, but that ancestry originates from many heterogeneous ancestors rather than one (or a few) homogeneous ancestor(s).

Our simulation results between A-East and A-West, as well as between more detailed ancestor groups, showed more variance in the ancestry estimates for individuals with more heterogenous background than for the individuals with ancestors from a single origin. We noticed that our estimation procedure always introduced small proportions of ancestry from each of the available reference groups. Based on this, we assume that small proportions of ancestry (~3–6%) should not be interpreted as a reliable evidence of direct ancestry in the reference group but rather as natural variation in a continuous measure. Shrinking away such small ancestry proportions, for example by thresholding at 5%, could improve the interpretability of the results, as was shown for the simulated results in S11 Fig. This, and other technical ways to improve the accuracy and interpretation of the results, warrant a more detailed study in the future.

SOURCEFIND is a software tool that works directly on the output of the haplotype-based methods ChromoPainter and FineSTRUCTURE[9]. It has been previously utilized to capture genetic ancestry within Europe [12, 16, 40] and in Latin America[18]. We have previously shown that ChromoPainter and FineSTRUCTURE work well in our target population[27]. Therefore, SOURCEFIND was a natural candidate for testing how well our reference groups identify genetic ancestry. Previously, SOURCEFIND has been shown to give highly concordant continental-level ancestry estimates with a standard ADMIXTURE-analysis[8] and to outperform a ChromoPainter-based NNLS method [18]. We are not aware of a direct comparison between SOURCEFIND and other haplotype-based methods, such as RFmix[41].

Individual's genetic ancestry has also been shown to crucially affect the interpretability of polygenic risk scores, that are currently heavily studied for medical genetics applications[42, 43]. The challenges of polygenic risk scores manifest not only between genetically distant, intercontinental populations[42, 44] but also between subpopulations within a country[45] demonstrating the need for characterization of genetic structure at finer scales. In fact,

incorporating the genetic ancestry information in polygenic risk scores has shown to improve the genetic risk estimation[1]. We consider this topic as an important future application area of the framework that we have presented here.

To conclude, this work demonstrates the power of ancestry estimation to reveal detailed demographic events as well as the continuous, gradually changing and mixing nature of real-world human populations. The work serves as a basis for detailed ancestry estimation within Finland, and we also provide a website for general public to interactively examine our results. We expect that our results help personalizing future genomic medicine in the Finnish population and promote the participation of general public in large-scale biobank collections that provide unprecedented opportunities for human genetic research, in Finland as well as elsewhere in the world.

# Materials and methods

## Data

Data originate from the National FINRISK Study, a population-based random sample of individuals aged 25–74 years[31]. The study has been conducted every five years since 1972. DNA was collected first time in 1992 study, and here we used individuals collected in years 1992, 1997, 2002, 2007 and 2012. In 1992, the study was conducted in four study areas; North Karelia and North Savo provinces in eastern Finland, Turku-Loimaa area in Southwestern Finland, and the cities Helsinki and Vantaa in southern Finland, in 1997 a fifth study area, provinces of North Ostrobothnia and Kainuu (in 2002 also the province of Lapland) in northern Finland, was included. Thus, the study covers different geographic areas of Finland well and nearly half of the Finnish population is living in the study area. For each individual, the data included information on their genotypes, birth place and birth year. For a large part of the individuals, the data also included their parents' birth place information. All study individuals had given a written consent.

## Data quality control

All individuals were genotyped with Illumina's HumanCoreExome genotyping chip. Starting from 23,431 individuals and 538,840 genetic variants, we first excluded variants with minor allele frequency (MAF) below 5%, Hardy-Weinberg equilibrium p-value below 1e-6 and variants that were missing in over 1% of the samples. We also excluded variants with duplicated genotype positions. These statistics were calculated using PLINK 1.9[46, 47]. After these filters, the data included 229,844 variants.

We excluded individual samples if their heterozygosity was below -0.04 or above 0.04 and if variant-missingness was above 0.005 calculated with PLINK 1.9[46, 47]. We calculated individual relatedness with KING 2.1.4[48] and excluded 3,635 individuals with a 3ʳᵈ degree relative (kinship over 0.0442) in the data. Further, we excluded the individuals who themselves or whose parent was born abroad (no Finnish birth municipality available), and the individuals on two bad-quality genotyping plates, identified in our previous study[27]. For the haplotype-based ancestry analyses, we also excluded the individuals with a high proportion (over 10%) of missing variants in chromosome 21. For the analyses using region of birth, we further excluded the individuals marked to have been born in the municipality of Karjala (municipality code 222), located in SWF, because it was impossible to know whether they were truly born in that municipality or rather in the larger region of Karelia that is in Finnish called with the same name (Karjala). Additionally, we ran PCA of our samples together with the non-Finnish samples from 1000 Genomes project[49]. By utilizing K-nearest neighbors method (K = 21), we excluded 31 individuals who showed closer relatedness to the 1000 Genomes samples than

to our Finnish reference candidates (S16 Fig). Finally, our data included 18,463 individuals with good-quality genotypes and location information. S5 Table shows the number of excluded or included samples after each filtering step.

## Identification and spatial sampling of reference candidates

To define geographically motivated reference groups, we first identified over 8,187 individuals whose parents were born within 80 km from each other and calculated their geographic location as the mean of their parents' birth places (available at the level of municipality). Most of these individuals were geographically clustered in either eastern or southwestern corners of Finland and because we had previously discovered that an uneven geographic sampling density can affect the identification of subpopulations[27], we applied the following spatial sampling procedure to thin down the individuals from the densest areas.

For each individual, we calculated how many neighbors they had in their immediate proximity (local neighbors, within 5 km) and in their wider proximity (global neighbors, within 30 km). Then, we excluded individuals in two steps. First, we found all individuals that had more than 15 local neighbors and randomly excluded one such individual as long as all individuals had 15 or less local neighbors. This reduced the number of samples in large cities and ensured more even sampling for the next step. Second, we found all individuals that had more than 40 global neighbors and, among them, we identified those who had the most local neighbors and randomly excluded one of those. This was repeated until no individual had more that 40 global neighbors. S2 Fig demonstrates the impact of the procedure on the geographic distribution of samples. The procedure resulted in 2,754 individuals from whom we further excluded 10 geographic outliers and 3 outliers in PCA on ChromoPainter's coancestry matrix as described in [9], resulting in 2,741 individuals. These 2,741 individuals were used as the donors in ChromoPainter analyses and they formed our set of reference candidates. The 5,446 individuals who were excluded during this spatial sampling procedure were later used as ancestor candidates in simulations to test the ancestry estimation.

## Chromosome painting with ChromoPainter

All individuals were first phased together using SHAPEIT2 software[50], 229,844 genetic variants, and an average European effective population size of 11,418 and the HapMap phase II recombination map. Then, both for identifying reference groups and for further estimating ancestry, we identified pairwise haplotype-sharing patterns, so called chromosome paintings, using ChromoPainter v 2.0[9]. ChromoPainter estimates the number and the cumulative length of haplotype blocks shared between the test individual and all donor individuals using dense genotype data. We ran ChromoPainter using the 2,741 reference candidates as donors, and the average switch rate (-n 3720.27) and global mutation probability (-M 0.00014), estimated over chromosomes 1, 9, 15 and 22, and with 28 donor individuals using 10 EM-iterations. Other parameters were kept at their default values.

## Identifying genetic populations with FineSTRUCTURE

To identify genetically homogeneous reference groups within the 2,741 geographically defined reference candidates, we ran FineSTRUCTURE 2.0.1[9]. First, we performed the Markov chain Monte Carlo (MCMC) run using 1,000,000 burn-in iterations and 2,000,000 sample iterations, saving every 10,000th iteration. The MCMC run was followed by an additional re-assignment where the population assignments were assessed against the individuals' maximum assignments during the MCMC iterations following Leslie et al. 2015[10]. Finally, the clusters were merged with the FineSTRUCTURE's tree-building method maintaining the tree order

and likelihoods. The procedure was repeated with a different seed for MCMC run to check the convergence.

## Ancestry estimation with SOURCEFIND

To estimate the haplotype-based ancestry within Finland, we used SOURCEFINDv2[18], a software specifically implemented to work with ChromoPainter. SOURCEFIND uses an averaged chromosome painting of the reference groups to find the ancestry proportions for the test individual/population utilizing an MCMC method. In our analyses, we used 50,000 burn-in iterations, 200,000 sample iterations and recorded the results from every 5,000th iteration. The copyvector, used to compare chromosome painting patterns between the test individual and the reference groups, was defined as an average over 20 FineSTRUCTURE populations (S3C Fig). We used 2, 6 and 10 reference groups (called surrogate groups in SOURCEFIND) to estimate ancestry. For individuals in the reference groups, ancestry was estimated by leaving the individual itself out from the reference group in the SOURCEFIND analysis. For other individuals, all reference individuals were used.

## Selection of ancestor candidates

To test the estimation of genetic ancestry within Finland, we simulated individuals using real data. We started by identifying ancestor candidates that would be geographically motivated similarly to our reference candidates. We reutilized those over 5,446 individuals who were excluded from the reference candidates in the spatial sampling procedure but whose parents were born within 80 km from each other.

For testing refset 2, we selected two ancestor candidate sets, one from West and one from East. The western ancestor candidates (A-West) were those whose parents were born either in Southwestern Finland (SWF) or in Ostrobothnia (OST) and the eastern ancestor candidates (A-East) had their parents from Ceded Karelia (CKA), North Karelia (NKA), Kainuu (KAI), North Ostrobothnia (NOS) or Lapland (LAP). The region of North Savo (NSA) was not included as it would have caused a significant imbalance in the geographic distribution of the ancestor candidates. The candidates were further filtered down based on PC1 coordinate: 80% of western (eastern) ancestor candidates with the smallest (largest) PC1 coordinate were kept (S8 Fig).

For testing refsets 6 and 10, we selected 7 groups of ancestor candidates in such a way that both of the parents were born in the same region, either in SWF, OST, LAP, KAI, NKA, CKA or in the municipalities of Kuusamo, Taivalkoski and Pudasjärvi of NOS (referred to as A-Kuusamo). In addition, for each of the 7 sets, we estimated a two-dimensional normal density function in PC1-PC2 space and kept the individuals who were within the highest 50% density region (S8 Fig). We did not use the genetic data of our reference individuals or SOURCEFIND estimates were not used in the selection process of the ancestor candidates.

## Principal component analyses

We performed principal component analyses both within the FINRISK data and together with the samples of the 1000 Genomes Project[49] using PLINK 1.9[46, 47].

Principal component analysis within the FINRISK samples was performed for 18,719 individuals and 56,661 LD-independent variants. These data included 256 individuals who were not part of the haplotype-based analyses as they were only later excluded due to ambiguous or missing location data or as outliers of the 1000 Genomes PCA (S16 Fig). LD-independent variants (56,661) were defined using command--indep-pairwise with 1500 kb window size, 500 kb

step size and 0.2 as r$^2$ threshold in PLINK 1.9, and by further excluding the long-range LD regions described in [51].

Principal component analyses together with the 1000 Genomes data were performed on 18,715 FINRISK samples and 1,536 non-Finnish samples of the 1000 Genomes phase 3 data using 49,423 LD-independent variants. We performed 6 separate PCA runs: one with all five super populations (314 Africans, 264 Americans, 480 East Asians, 380 Europeans, and 98 South Asians) together with the FINIRSK samples, and also 5 runs, where each super population was separately analyzed with the FINRISK samples (S16 Fig).

## Simulating individuals from ancestor candidates

For each simulation scenario, we simulated 20 individuals by first randomly sampling ancestors from the ancestor candidate groups (S9 Fig) and then simulating recombination between the ancestor haplotypes. Each ancestor candidate was used only once in one simulation run. We simulated new haplotypes for chromosomes 1–22 in R (see Data Availability Statement) by first sampling crossing-over events within the ancestors' phased chromosomes and then sampling one new haplotype from each ancestor. The crossing-over events were sampled between two loci according to the probabilities from the HapMap phase II recombination map. When simulating several generations, we generated new haplotypes iteratively by starting from the eldest ancestors (S7 Fig).

## Local regression for ancestry curves

We estimated the average ancestry proportions for each region by using local regression method, LOESS (locally estimated scatterplot smoothing), in R[52]. We regressed the individuals' ancestry profiles on individuals' birth years by fitting LOESS using value 0.5 for the spatial smoothing parameter.

## Changes in genetic heterogeneity

We estimated genetic heterogeneity of ancestry profiles using entropy measure to quantify the change in heterogeneity over time. For each region and for each yearly ancestry profile averaged over individuals born in that year, we measured entropy as

$$H = - \sum_{i=1}^{k} (p_i \, log_2 \, (p_i))$$

where $p_i$ is the proportion of the ancestry profile assigned to reference group i = 1,...,$k$, where the number of reference groups in profile was $k$ = 2, 6 or 10. Instead of comparing the absolute values of entropy that can depend strongly on the reference groups, we quantified the change by predicting entropy with birth year by a simple linear regression where each yearly observation was weighted by the number of samples used to calculate the yearly ancestry profile. The slope of the model is then interpreted as the rate of change in genetic heterogeneity, negative values indicating a decrease in heterogeneity and positive values indicating diversifying profiles. We estimated the rate of change for the whole time period from 1923 to 1987, as well as separately for the periods before and after the year 1950, corresponding to the period including the Second World War and the post-war period, respectively. S21 Fig shows the yearly entropy and the fitted model for each region with refset 10.

## Average inbreeding coefficient, F

To complement the measure of change in heterogeneity of the ancestry profile per region, we also computed the average inbreeding coefficient, F, for each study region. The individual

inbreeding coefficients were first calculated using PLINK 1.9[46, 47] and then averaged over the individuals born in the region.

## Pairwise-$F_{ST}$

We calculated pairwise-$F_{ST}$ between the reference groups (Fig 2) and the ancestor candidate groups (S9 Fig) using SmartPCA of EIGENSOFT package[7] (fstonly: YES, fsthiprecision: YES) and 56,661 LD-independent variants.

## Average ancestry profiles over municipalities

In an interactive web application, we present the average ancestry profiles for the Finnish municipalities. These profiles have been estimated by weighting the individual profiles by the inverse of a squared distance between the individual and the center of the municipality as

$$A_m = \frac{1}{r_{Tot}} \sum_{i=1}^{N} \frac{a_i}{r_{im}^2},$$

where $A_m$ is the ancestry profile of a municipality m, $a_i$ is the ancestry profile of an individual i, $r_{im}$ is the distance between the birth place of individual i and the municipality m, $r_{Tot}$ is the sum over $1/r_{im}^2$. Thus, the ancestry profile of a municipality does not represent the average profile of only the individuals born in that municipality but may also be affected by the individuals from the neighboring municipalities. The minimum distance between an individual and a municipality was set to 5 kilometers to avoid very high weights in the calculation.

## Maps

Maps in figures were generated by using data from geoBoundaries[53].

## Supporting information

**S1 Fig. Workflow of the study.**
(TIF)

**S2 Fig. Geographic distribution of study samples chosen by parents' birth places.** Geographic distribution of samples whose parents were born within 80 km from each other A) before the spatial sampling procedure, B) after excluding the individuals with a high number of local neighbors and C) after excluding the individuals with a high number of global neighbors. The individuals are displayed at the mean of their parents' municipalities of birth after adding some jitter to the points to ensure the anonymity. Geographic outliers are excluded.
(TIF)

**S3 Fig. FineSTRUCTURE tree and populations.** FineSTRUCTURE tree at level 20 and the corresponding populations on a map when the tree is cut at A) level 2, B) level 15 and C) level 20.
(TIF)

**S4 Fig. Geographic outliers.** Maps shows the location of individuals excluded as geographic outliers of A) refset 2 (3 individuals), B) refset 6 (11 individuals) and C) refset 10 (16 individuals). The excluded outliers are highlighted with black X-marks. The included individuals are shown with pale colors.
(TIF)

**S5 Fig. Identity proportions for the FineSTRUCTURE populations.** Panel A) shows the identity proportions when all 15 population were used as reference populations. Panel B) shows the identity proportions using only the 10 populations that show identity proportion above 0.50 in panel A. Colors correspond to the populations in S3B Fig.
(TIF)

**S6 Fig. Age distributions of reference groups and ancestor candidates.** Age distributions of the reference groups of A) refset 2, B) refset 6, C) refset 10 and D) the ancestor candidates. The boxplot whiskers show the range, the boxes show the interquartile range and the dark line shows the median of the birth years.
(TIF)

**S7 Fig. Schematic representation of our simulation strategy.** In each simulation, $2^G$ individuals were sampled to represent the ancestors from G generations back in time (black box), where G varied between 1 and 5. All the subsequent descendants in generations G-1, G-2, ..., were simulated to determine the genotypes of the target individual at generation 0 (grey box). In this example simulation, 1 ancestor is sampled from A-West (red) and the remaining $2^G - 1$ ancestors were sampled from A-East (blue). The two adjacent bars correspond to the two haplotypes of an individual and the color corresponds to the ancestor candidate group.
(TIF)

**S8 Fig. Location of ancestor candidates on genetic principal component space.** The location of ancestor candidates on a plane defined by principal components (PC) 1 and 2 of the genetic structure for A) simulation settings for refset 2 and B) simulation settings for refsets 6 and 10.
(TIF)

**S9 Fig. Geographic location of ancestor candidates.** The geographic location of the ancestor candidates in simulation settings A) for refset 2 and B) for refsets 6 and 10. The names of the ancestor candidate groups are shown on right.
(TIF)

**S10 Fig. Individual ancestry profiles for simulation scenarios between East and West.** Ancestry profiles for 20 individuals in simulation settings involving ancestry groups R2_*East* and R2_*West*: A) for setting *All-West*, B) for *All-East*, C) for *Almost-East* and D) for *Almost-West*. Blue denotes the estimated proportion in reference group R2_East and red denotes the proportion in reference group R2_West. Gen (1,...,5) refers to the number of generations considered in the simulation. 'Mean' shows the average over the 20 simulated individuals.
(TIF)

**S11 Fig. Average ancestry profiles for detailed simulation scenarios without below 5% proportions.** Average ancestry for the simulation results in Fig 4 when the ancestry proportions below 5% were shrunk to zero and the remaining proportions were scaled to one. Panel A) presents individuals whose all ancestors come from one group (single origin) shown in the title estimated using refsets 2, 6 or 10. Panel B) presents individuals whose $2^G$-1 ancestors, where G = 1...4 is the number of generations, originate from A-Southwest and 1 ancestor originates from the ancestor group in the title, estimated using refset 10. The colors correspond to the reference groups in Fig 2.
(TIF)

**S12 Fig. Individual ancestry profiles for detailed single origin simulation scenarios.** The individual ancestry profiles for 20 individuals whose both parents originate from the ancestor candidate group of A) A-Southwest, B) A-Bothnia, C) A-N_Karelia, D) A-Kainuu, E)

A-Kuusamo, F) A-Lapland and G) A-Evacuated (see Fig 4A for the mean values). The colors correspond to the reference groups in Fig 2.
(TIF)

**S13 Fig. Average ancestry profiles for detailed simulation scenarios.** Detailed simulation results for mixed ancestry from ancestor groups A-Southwest and A-N_Karelia. Panel A) presents individuals whose $2^G$-1, where G is the number of generations, ancestors originate from A-Southwest and 1 ancestor originates from the region in the title. Top row shows the ancestry profiles estimated using refset 2, middle row shows the same for refset 6 and bottom row shows them for refset 10. Panel B) shows the same quantities for a simulation setting where all but one ancestors originate from A-N_Karelia.
(TIF)

**S14 Fig. Individual estimates of the major ancestry component in detailed simulation results.** Detailed simulation results for mixed ancestry from ancestor groups A-Southwest and A-N_Karelia (corresponding to S13 Fig). Panel A) presents individuals whose $2^G$-1, where G is the number of generations, ancestors originate from A-Southwest and 1 ancestor originates from the region in the title. Top row shows estimated ancestry in R2-West, middle row shows the same for R6-Southwest and bottom row shows them for R10-Southwest. Panel B) shows the same quantities for a simulation setting where all but one ancestors originate from A-N_Karelia and the reference groups whose estimates are shown are R2-East (top), R6-Savo-Karelia (middle) and R10-Savo-Karelia (bottom).
(TIF)

**S15 Fig. Ancestry profiles for simulated first-generation mixed individuals.** Average ancestry profiles for 20 simulated individuals whose parents come from different geographic regions are shown. Title describes the ancestor candidate groups of the parents used in the simulation.
(TIF)

**S16 Fig. Principal component analyses with 1000 Genomes samples and FINRISK samples.** A) PCA of 5 super populations of the 1000 Genomes (Phase 3) samples and our Finnish FIN-RISK (FIN-FR) samples. PCA of the FINRISK samples together with the B) non-Finnish European (EUR), C) East Asian (EAS), D) American (AMR), E) African (AFR) and F) South Asian (SAS) samples of the 1000 Genomes Phase 3. The FINRISK samples circled with red were identified to show admixture with one or more super populations and were excluded from the regional ancestry analyses. None of our reference individuals was among the excluded.
(TIF)

**S17 Fig. Development of genetic ancestry profile in 12 regions using refset 2.** The map on the left shows the regions and the locations of the individuals who form the 2 reference groups. The curves show the estimated ancestry proportion in each reference group as a function of the birth years of individuals born in each region (name of the region in the title).
(TIF)

**S18 Fig. Development of genetic ancestry profile in 12 regions using refset 6.** The map on the left shows the regions and the locations of the individuals who form the 6 reference groups. The curves show the estimated ancestry proportion in each reference group as a function of the birth years of individuals born in each region (name of the region in the title).
(TIF)

**S19 Fig. Development of genetic ancestry profile in 12 regions using refset 10.** The map on the left shows the regions and the locations of the individuals who form the 10 reference

groups. The curves show the estimated ancestry proportion in each reference group as a function of the birth years of individuals born in each region (name of the region in the title).
(TIF)

**S20 Fig. Regional changes in genetic ancestry from the R10-Evaluated reference group.** Changes in the genetic ancestry proportion within mainland Finland using refset 10 but showing only the ancestry proportion from R10-Evacuated. The map on the left shows the location of reference individuals and R10-Evacuated is located at southeast corner of the map overlapping the region of Ceded Karelia (CKA).
(TIF)

**S21 Fig. Entropy values regressed on birth years for the 12 study regions.** The points represent entropy of the mean profile of individuals born during one year in a particular region estimated with respect to refset 10. The size of the point represents the number of individuals averaged in that yearly profile. The purple line is the linear regression line fitted to the data. The slope estimates the rate of change in heterogeneity of the ancestry profile.
(TIF)

**S22 Fig. Rate of change in heterogeneity at each study region before (1923–1950) and after (1951–1987) the year 1950.** The rate of change is calculated by regressing the mean entropy over the years with refsets 2, 6, and 10 on the year of birth.
(TIF)

**S23 Fig. Average inbreeding coefficient, F, per study region.** The values at the bottom report the average F per region with its standard error in parentheses. Whiskers show the 95% confidence interval.
(TIF)

**S1 Table. Number of incorrectly assigned individuals in simulations between East and West.** The number of individuals incorrectly assigned to a single origin, out of 20, based on whether their A) West ancestry component or B) East ancestry component was above the threshold. The threshold was defined as the second largest value in A) *All-West* or B) *All-East* simulation setting (corresponding to the 95% quantile in the simulation setting).
(PDF)

**S2 Table. Pairwise-FST values ($\times 10^5$) between ancestor candidate groups and reference groups of refset 10 (lower triangular) and their standard errors (upper triangular).**
(PDF)

**S3 Table. Ancestry proportions for single-origin simulations categorized into expected and unexpected ancestry.** Column 'Expected ancestry' shows which reference groups were considered the closest to the corresponding ancestor group, and the 'Total expected ancestry' sums over the expected ancestries. Unexpected ancestry was defined as everything else except the expected and the average contribution of those groups is shown in column 'Average unexpected ancestry'. Column 'No shrink' shows the values of the raw ancestry estimates and column 'Shrink <5%' shows the results after shrinking the individual ancestry estimates below 5% to zero and rescaling the remaining non-zero ancestry proportions back to 100%.
(PDF)

**S4 Table. Range of the birth years (Min and Max) and the mean birth years of the reference groups.** Mann-Whitney p-value corresponds to a test between the focal group and the union of the rest of the groups at that refset.
(PDF)

**S5 Table. Number of study individuals excluded and included after each filtering step.**
(PDF)

**S6 Table. Number of reference candidates excluded and included after the steps of reference group identification process.** The numbers in parentheses refer to the number of populations excluded or included.
(PDF)

## Acknowledgments

The data used for the research were obtained from THL Biobank under the project BB2019_44. We thank all study participants for their generous participation at THL Biobank and the National FINRISK study.

## Author Contributions

**Conceptualization:** Sini Kerminen, Samuli Ripatti, Matti Pirinen.

**Data curation:** Sini Kerminen.

**Formal analysis:** Sini Kerminen.

**Funding acquisition:** Samuli Ripatti, Matti Pirinen.

**Investigation:** Sini Kerminen.

**Methodology:** Sini Kerminen, Matti Pirinen.

**Project administration:** Matti Pirinen.

**Resources:** Aki S. Havulinna, Markus Perola, Pekka Jousilahti, Veikko Salomaa, Mark J. Daly.

**Supervision:** Samuli Ripatti, Matti Pirinen.

**Visualization:** Sini Kerminen, Nicola Cerioli, Darius Pacauskas, Rupesh Vyas.

**Writing – original draft:** Sini Kerminen, Matti Pirinen.

**Writing – review & editing:** Sini Kerminen, Nicola Cerioli, Darius Pacauskas, Aki S. Havulinna, Markus Perola, Pekka Jousilahti, Veikko Salomaa, Mark J. Daly, Rupesh Vyas, Samuli Ripatti, Matti Pirinen.

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
