## [Decision Letter · Decision Letter 0]

28 Sep 2020

Dear Dr Pirinen,

Thank you very much for submitting your Research Article entitled 'Changes in the fine-scale genetic structure of Finland through the 20th century' to PLOS Genetics. Your manuscript was fully evaluated at the editorial level and by independent peer reviewers. The reviewers appreciated the attention to an important topic but identified some aspects of the manuscript that should be improved.

We therefore ask you to modify the manuscript according to the review recommendations before we can consider your manuscript for acceptance. Your revisions should address the specific points made by each reviewer.

[LINK]

Yours sincerely,

Simon Gravel, Ph.D.

Guest Editor

PLOS Genetics

Hua Tang

Section Editor: Natural Variation

PLOS Genetics

Both reviewers appreciated the attention to an important problem, and the relevance of the analyses.

Both reviewers also expressed some confusion about the clustering and filtering steps, and I think that this would be important to address. Perhaps a graphical representation, as suggested by Reviewer 2, would be helpful.

I would find it also important to report the number of participants excluded through each filtering step. E.g., on line 206, the manuscript lists the number of outliers, but not the proportion of individuals excluded for low identity proportions. This is important to ascertain how representative the analysis is the the actual population of Finland (see also point 2 of reviewer 2).

On line 241, it was unclear to me how the candidate ancestors were selected, in particular whether they were selected after the filtering steps (in which case the ancestry analysis would be over-confident, since the simulations used individuals that cluster exceptionally well.)

Figure 4 shows average ancestry over multiple simulated individuals. As I understand things, this would provide an estimate of the systematic bias in assignment, which is a relevant metric for the time-dependence analysis, but not of the uncertainty in assignment. I think this could be made clearer in the discussion of the results.

For all these reasons, the conclusions on lines 342-347 seem to overstate the accuracy of the regional inferences at the individual level. This is particularly important given that the manuscript highlighs forensics as a possible application of this type of research.

Minor points:

“homogeneous” and “distinct” reference groups – this has not quite been shown, and I would expect that a better powered or more detailed study would reveal that these groups are neither quite homogeneous nor quite distinct. Given that this work has a public outreach component, I would advocate for more careful language given how humans like to overinterpret genetic differences across groups.

This is especially true here since the approach used extensive filtering to reach the “homogeneous” groups, and therefore the figures (such as Fig. 2) give an exaggerated idea of the divergence between populations.

Similarly, I would encourage the authors to avoid expressions such as “genetically intact”, which suggest a positive connotation to lack of mixing. (e.g., “genetically isolated” would be preferable).

Reviewer's Responses to Questions

**Comments to the Authors:**

Reviewer #1: This is an excellent paper that I thoroughly enjoyed reading. The methodology is sound and is in-line with the state-of-the-art in ancestry estimation. The results are very interesting, and the level of detail and precision on population movements within Finland are unprecedented. It is well-written, and well structured. I have only minor comments on the exposition to offer.

- My only criticism of this paper is that I found the terminology around the three "levels" of detail in which the population was studied very confusing. This made sense when discussed in the introduction, when talking about 2, 6 and 10 reference groups of ancestry. However, when you speak later of (say, line 188) about level-2 reference groups, and then around line 203 of level 10 and level 6, I thought you were talking about different heights in the FS-tree. It became clearer later on what you mean when I read the methods, but I think the terminology could be a lot simpler and clearer here. Maybe use 2-way, 6-way and 10-way instead? This gives a better intuition to the reader that it's about splitting into groups, and "level" just seems misleading. I think "level" is being used in a few different ways, so it would be good to go through the text with this in mind.

- Line 68: "A variety of methods"

- Line 75: The term "British Isles" is politically charged and regarded as offensive by many Irish people (please see the wikipedia page for a good summary: https://en.wikipedia.org/wiki/British_Isles_naming_dispute). Please consider avoiding this contentious terminology by using a more neutral term such as "Britain and Ireland".

- Line 87. Delete "also"

- Line 91 (and elsewhere). Should it be "the Soviet Union"?

- Line 197: rerun -> reran

- Line 337. This should be 2^{G - 1}, shouldn't it?

- Line 489. I couldn't parse this sentence - what is it trying to say?

- Line 497. Swap "rather continuous" to "continuous rather".

Reviewer #2: In this paper, Kerminen and collaborators use state-of the-art population genetics tools to investigate the population structure of Finland, specifically to see how major events in the 20th century affected this structure. FineSTRUCTURE was used to partition a reference population into discrete subgroups, while SOURCEFIND was used to estimate the proportion of ancestry from each subgroup for each individual in a testing population, using entropy measurement and year of birth data to quantify the changes in ancestry over time.

The authors’ findings matched what is known from the historical and demographic record, and further contributed that the migration of people from regions of Finland that were annexed by the USSR (and continue to be part of Russia to this day) have had the largest detectable effect on the population structure of Finland. Events such as the urbanization of the population in the latter half of the 20th century show much smaller, more local effects. Additionally, the study demonstrates the technical limits of SOURCEFIND to distinguish reliably ancestry proportions < ~5% from background population genetic variation. This has implications for inference of contributions further back in an individual’s lineage.

Generally, I think the work done is of excellent quality and that the conclusions are supported by the results shown. I have three main comments on the methodology, and several minor suggestions:

Main comments:

1.Date of birth of “reference candidates”.

Given the question asked, I would have thought that choosing the “reference candidates” based on earliest year of birth would have made sense, so that these reference groups really reflect the genetic background of the reference group early in the 20th century. I am not sure why the author decided not to choose reference individuals based on this info (l.514-518). In the >8000 potential candidates they reported, they made sure that they had good coverage across the country, but I feel that they could have consider selecting the people with the earliest date of birth as well. My concern is that, if a certain reference group is biased towards early ancestors (as reported l.516) and the other is biased towards later ancestors, this could potentially lead to strange effects when looking at admixture proportions in the regional subgroups. Similarly, were the ancestors in the simulations selected based on a logical date of birth scheme (ie. first ancestors are the oldest, with G1>G2>G3>G4)? If not, what is the expected impact of overlooking this aspect (that needs to happen in reality) on their results?

2. Admixture from outside Finland.

The authors mention in discussion the fact that the individuals could could have had ancestors from outside of Finland, and I am not really sure what the impact of that could be on the results presented, especially if that distribution of admixture is uneven between reference candidate or the tested regional subgroups. A solution would have been to "masked out" the chunks from distant ancestry in the genomes of individuals. And even leaving out recent immigration to Finland from countries all over the world, there has always been gene flow between Finland and Sweden (also probably Russia?). Could the authors show that these kind of admixture event would not (or only negligibly) bias their results?

3. SOURCEFIND

I did not know much about SOURCEFIND before reading this manuscript and wish that there had been more justification for why it was used over, for instance, applying RFMix or other alternative methods? I would have liked to see more discussion of the implications of the fact that SOURCEFIND only seems to make accurate inferences about recent ancestry. How does this compare to other software? If someone were interested in making inferences about more remote ancestors, is there any existing software suitable to that question? How much of SOURCEFIND’s uncertainty is a function of the specific population history of Finland? Would it be more or less accurate in a more heterogeneous population?

Minor suggestions:

- I had several questions on the simulation strategy while reading the results and I think that a figure, showing the simulation scheme graphically, would be beneficial to the reader. For example, it could clarify the fact that in the Almost-East/West simulations, the foreign ancestor was always drawn as a G1 ancestor (I think? from the results in Figure 3.. although I am not sure because legend says "a single ancestor, G generations back in time" l.271, suggesting it could be any G value?)

- I am a bit confused with the wording "location of individuals" - does it refer to birth place, or where these individuals live/were sampled (eg. Figure 2)? Similarly, "parents’ geographic location" and "parental birthplace information" are used... I think this wording should be classified throughout.

Similarly, I was a bit confused by the term "newborns" in several places (l.410,458,488) - what does this mean?

- l. 647 "For individuals in the reference groups, the ancestry was estimated by leaving the individual itself out from the reference group." I am not really sure I understand how this is done in practice? Is FineSTRUCTURE rerun on the entire dataset by leaving one individual out?

- l.432-433. absolute genetic diversity in regions is defined in opposition to average heterozygosity. I was curious as to whether average heterozygocity in the different regions has been computed as well, and if there are notable differences between regions or if it is quite homogeneous (maybe this has been done in a previous study, if so, please cite).

- In my opinion, the term "admixture" is generally refering more to the mixing of genetic material from a distantly-related populations. I dont know what a better term would be though (gene flow? genetic mixing?)... or maybe it could simply be explicitely defined in the introduction that the terms admixed/admixture (normally used for more distantly related populations) will refer here to genetic mixing from/ gene flow between closely related populations.

Website:

The website is great and very appealing! However, on Safari I see a truncated version (the right side goes outside the page and we can't slide the page - at least on two different computers). Also, on iPhone, the ancestry/tree panel is not displayed at all (might be too heavy for mobile - but just wanted to report it).

Typos

- l.140-142 "We will first introduce a procedure to identify suitable reference groups, then we test their performance to detect ancestry via simulations, and finally apply them to estimate the ancestry of 18,494 FINRISK samples to characterize" >> I am not sure what "their performance", "them" refers too, probably to the methods?

- l.588 "For birth region analyses, we further excluded individuals marked to had been born in the municipality of Karjala" >> ... to have been born ...

- l.708 "We estimated the rate of change for whole time period from 1923 to 1987" >> ... for the whole time period ...

- Some figures are missing labels on axes (eg. Figures 3,4, S7-10)

**Have all data underlying the figures and results presented in the manuscript been provided?**

Reviewer #1: Yes

Reviewer #2: Yes

PLOS authors have the option to publish the peer review history of their article (what does this mean?). If published, this will include your full peer review and any attached files.

Reviewer #1: **Yes: **Jerome Kelleher

Reviewer #2: **Yes: **Julie Hussin

---

## [Decision Letter · Decision Letter 1]

15 Dec 2020

Dear Dr Pirinen,

Thank you very much for submitting your Research Article entitled 'Changes in the fine-scale genetic structure of Finland through the 20th century' to PLOS Genetics.

The manuscript was fully evaluated at the editorial level and by independent peer reviewers. The reviewers appreciated the attention to an important topic but identified some concerns that we ask you address in a revised manuscript.

Most of these concerns are requests and suggestions for clarification. We therefore ask you to take these into account in submitting a revised manuscript. 

[LINK]

Yours sincerely,

Simon Gravel, Ph.D.

Guest Editor

PLOS Genetics

Hua Tang

Section Editor: Natural Variation

PLOS Genetics

Reviewer's Responses to Questions

**Comments to the Authors:**

Reviewer #1: The authors have addressed all of my points well, I have no further comments.

Reviewer #2: I thank the authors for answering my questions and addressing them in the manuscript. I think the changes they made have greatly improved the readability of the paper and have made it more comprehensible.

I have two final minor comments:

- Line 448: “whence” means “from which” or “from where”. I think the authors mean “where” or "for which" here.

- On the issue of the word “newborns” : When this term is used, it’s usually referring to people who are currently newborns or to discuss some facet of infancy where being newborn is relevant (for example, “newborns can’t focus their eyes”). It’s not generally used the way the authors have used it in this paper. I would use the clunkier, but more accurate “individuals born in 19XX” in its place.

Reviewer #3: This paper by Kerminen and al. is simultaneously addressing the question of genetic structure of Finland, which is not totally new and the properties of SOURCEFIND algorithm when the source populations show small differentiation (at the fine greographical scale, within one country, be it Finland).

Revisiting the genetic structure of Finland and brings a very interesting approach, the analysis of rapid change in time of this structure, because of dramatic events. In this regard, the approach is original and worth reporting.

The introduction/title could provide more clear description of the study’s goal. This work basically tests, using simulation from realistic data, whether SOURCEFIND can correctly identify origin when the source population display only limited difference.

Moreover, in my opinion, this introduction should stress earlier one of the original points, the possibility to stratify the changes in genetic composition within a short period of time.

I don’t find very clear the references of the use of Globetrotter and Sourcefind. The message seems to be that the methods have been applied to populations that are large whereas in this study it is going to focus on fine-scale structure of a supposedly less broad populations. However, the referenced studies focus for instance on Ireland. I guess here it would be more clear to explictely state that you are testing these methods in a fine-scale context where source populations and target populations are very close. And where source populations are not heterogenous.

In terms of novelty the description of the genetic structure of Finland in present times has already been adressed and I find that they don't separate enough, even in the second part, what is really new from what is not. I think that the contribution of refugees from Soviet Karelia is new. Also, the evolution of the genetic structure in a short period of time is something important - actually, this is in my opinion the most important point. This is also quite new compared to previous papers where we could only estimate the change in population size from current genomes in the different clusters.

Concerning the simulations, I find it very interesting to start from existing chromosomes representative of a region in order to see Globetrotter's ability to estimate the proportions of origin of each population. They start from existing chromosomes (estimated in any case) as "founders" and simulate transmission. Somewhat in the spirit of HapGen which was distributed with the 1000 genomes. The "critics" asking to take founders with an older date of birth miss the fact that the problem is just theoretical: let's take ancestors representing populations with an Fst close to the classical Fsts between provinces.

The simulations seem to me to be valid in relation to the question posed. One thing has not been accounted for, however. This is the fact that they only capture a fraction of the haplotype diversity because they take the chromosomes as they are and observed whereas one could imagine generating generating founder haplotypes using the observed “source population” haplotypes but allowing for recombination at this stage – in the ancestors. Thus, they could have captured a wider and more accurate haplotype diversity while still relying on the observed structure. This is however a limited criticism as it is still of matter of simulating given a Fst and testing the consequences in lower generations. The simulation process (including the algorithm to identify “seed” founder populations) is more clear in this reviewed versions, as asked by editor and reviewers in the first round.

This paper (which seems to me very good and very pro) seems to chase two hares at the same time ... this is what is a bit annoying because it mixes a practical problem (structure and history of Finland) and a theoretical one - which uses very realistic (because real) data and therefore in a context - structure Finland.

The problem is interesting because it seems to me that Globetrotter seems to have been made to find admixture (and date it) from much more differentiated source populations. So it is and see the properties of the method to the extreme.

Even if this paper is a bit confusing because it piles up two (nearly three) topics, it has this novelty of following genetic structures on several generations and therefore comparing the impacts of internal migration (in the sense of the same people) and urbanization. Results from simulations can also give useful guidelines for interpreting SOURCEFIND results from other populations.

**Have all data underlying the figures and results presented in the manuscript been provided?**

Reviewer #1: Yes

Reviewer #2: Yes

Reviewer #3: None

PLOS authors have the option to publish the peer review history of their article (what does this mean?). If published, this will include your full peer review and any attached files.

Reviewer #1: No

Reviewer #2: **Yes: **Julie Hussin

Reviewer #3: No

---

## [Editor Report · Decision Letter 2]

6 Jan 2021

Dear Dr Pirinen,

We are pleased to inform you that your manuscript entitled "Changes in the fine-scale genetic structure of Finland through the 20th century" has been editorially accepted for publication in PLOS Genetics. Congratulations!

Yours sincerely,

Simon Gravel, Ph.D.

Guest Editor

PLOS Genetics

Hua Tang

Section Editor: Natural Variation

PLOS Genetics

Comments from the reviewers (if applicable):

**Data Deposition**

http://datadryad.org/submit?journalID=pgenetics&manu=PGENETICS-D-20-01240R2

**Press Queries**

---

## [Editor Report · Acceptance letter]

11 Feb 2021

PGENETICS-D-20-01240R2 

Changes in the fine-scale genetic structure of Finland through the 20th century 

Dear Dr Pirinen, 

We are pleased to inform you that your manuscript entitled "Changes in the fine-scale genetic structure of Finland through the 20th century" has been formally accepted for publication in PLOS Genetics! Your manuscript is now with our production department and you will be notified of the publication date in due course.

With kind regards,

Alice Ellingham

PLOS Genetics

On behalf of:
